# Effects of Reinforcement Ratios and Sintering Temperatures on the Mechanical Properties of Titanium Nitride/Nickel Composites

**DOI:** 10.3390/ma13204473

**Published:** 2020-10-09

**Authors:** Yi-Cheng Chen, Shih-Fu Ou

**Affiliations:** Department of Mold and Die Engineering, National Kaohsiung University of Science and Technology, Kaohsiung 807, Taiwan; 1105406101@nkust.edu.tw

**Keywords:** titanium, nickel, composite, sintering

## Abstract

In this study, powder metallurgy was used to fabricate titanium nitride/nickel metal-matrix composites. First, titanium and nickel powders with weight ratios of 20:80, 50:50 and 80:20 were dry mixed for 24 h. After cold isostatic pressing, the green compacts were soaked in a water-based hot forging lubricant and sintered at 850, 950 and 1050 °C for 1.5 h in an air atmosphere. The effects of the amounts of titanium powder and the sintering temperatures on the mechanical properties (hardness, wear resistance and compressive strength) of the composites were investigated. The results indicated that titanium gradually transformed into titanium nitride near the surface after sintering due to the carbothermal reduction reaction; this transformation was observed to significantly increase the hardness. In addition, an oxygen-rich film was observed to form between the titanium nitride particles and the nickel matrix. An optimum sintering temperature of 950 °C provides the composites (titanium–nickel weight ratios of 20:80) the best mechanical properties (wear resistance and compressive strength) among other groups. Furthermore, increasing the titanium content to 80% in the composite increased the hardness; however, the wear resistance and compressive strength decreased.

## 1. Introduction

Particle-reinforced metal matrix composites (PMMCs) are widely used in numerous applications such as in aviation, transportation, microelectronics and nuclear industries; this is because of their excellent specific strength, thermal conductivity, and high-temperature as well as abrasion resistances [1,2,3,4]. PMMCs are reinforced by ceramic particles that have been dispersed in a metal matrix. Their excellent mechanical properties are attributed to (1) an external force load transfer between the matrix and the reinforcement, (2) dislocation strengthening, (3) refined grain strengthening, (4) precipitation hardening, (5) solid solution strengthening, (6) mixed strengthening, and (7) synergistic strengthening [5]. The four types of methods generally used to manufacture PMMCs are stir casting [6,7,8,9], pressure penetration [10], powder metallurgy [11,12,13,14] and mechanical alloying [10]. Although stir casting is a low-cost method that has been employed worldwide [2], powder metallurgy can avoid the following unwanted phenomenon, namely: (i) agglomeration of the ceramic particles during mechanical agitation, (ii) settling of the ceramic particulates, (iii) segregation of the secondary phases in the metal matrix, (iv) extensive interfacial reactions, and (v) ceramic particulate fracture during mechanical agitation [15].

Nickel and its alloys are widely used in engineering applications in corrosion environments, such as chemical plants and nuclear power plants, due to their strength and corrosion- and wear-resistance at high temperatures. Such excellent properties make nickel of great interest for the choice of components in aggressive environments. In order to enlarge the range of nickel alloys’ applications in engineering fields, the nickel matrix composites produced by the addition of metal oxides, carbides and nitrides into the Ni matrix have been developed [16,17]. Titanium nitride has a high melting point (2927 °C), with high hardness and corrosion resistance and good thermal stability [18]. Titanium nitride has been used in tribological applications in different forms, i.e., thin film on machining tools [19,20] and reinforced particles in composites [21].

Ibrahim et al. produced titanium nitride/nickel composites via a direct current electrodeposition method. The results indicated that the titanium nitride microparticles in the nickel matrix greatly improved the corrosion resistance in a 3.5% NaCl solution and possessed higher hardness than a pure nickel [21]. Ramesh Bapu et al. synthesized titanium carbo nitride/nickel composites by electrodeposition, and evaluated the corrosion resistances of the composites. It was found that the titanium carbo nitride/nickel composites showed better corrosion resistance in 3.5 wt.% NaCl solution, and a higher hardness and better wear resistance than nickel. The degree of the improvement depends on the grain size and the volume percent of titanium carbo nitride in the composites [22].

Ceramic particles such as SiC, BN, TiC, TiB_2_, TiN, ZrO_2_, ZrN, MoC, WC and Al_2_O_3_ are commonly used as metal matrix composite reinforcements [3,10,23]. However, ceramic powders with diameters of less than 50 μm (d_50_ < 50 μm) are cohesive because of the large interparticle force (electrostatic, Van der Waal’s and liquid bridge forces) [24]. As such, achieving a uniform ceramic particle distribution in a metal matrix is challenging. Additionally, as the volume percentage of the reinforcing particles increases, agglomeration is more likely to occur [15]. The production of PMMCs with ceramic particles can be easily achieved by powder metallurgy. However, the cost of ceramic powder production is very high. This study aimed to use a carbothermal reduction reaction to transform metal particles into ceramic particles during sintering. In addition, the use of metal particles is expected to overcome the agglomeration of reinforcement particles in the PMMCs.

Herein, two kinds of metal powders were mixed first, namely titanium and nickel. The green compacts were soaked in a water-based hot forging lubricant and were sintered at elevated temperatures in an air atmosphere. The carbothermal reduction and diffusion reactions led to the formation of titanium nitride particles dispersed in the nickel matrix, and simultaneously densified the green compacts. The effects of the sintering temperature and the reinforcement ratio on the microstructure and mechanical properties of the composites were investigated. In addition, the interfacial adhesion between the metal matrix and the reinforcement particles was explored with wear tests.

## 2. Material and Methods

The mechanical properties of the composites strongly depend on the weight ratio of reinforcement particles and matrix materials. In this study, three weight ratios of reinforcement particles and matrix materials were chosen to find the optimal parameter in terms of the wear resistance and compressive strength of the composites. Furthermore, the sintering temperature significantly affects the density of the composites and, therefore, three sintering temperatures were applied.

### 2.1. Specimen Preparation

Commercial titanium (>99.5%) and nickel (>99.5%) powders with average diameters of 75 and 45 μm in spherical shape, respectively, were used. Titanium and nickel powders with three weight ratios of 20:80, 50:50 and 80:20 were dry mixed for 24 h with a V-shaped mixer. The powder mixtures were consolidated into cylindrical compacts (φ10 × 15 mm) by cold pressing at 25 °C with a pressure of 637 MPa under atmospheric conditions via universal testing machine (SHIMADUZ, UH-1, Kyoto, Japan). After soaking the green compacts for 30 s in a water-based hot forging lubricant, the green compacts were sintered at 850, 950 and 1050 °C for 1.5 h in air. The compacts with titanium/nickel weight ratios of 20:80, 50:50 and 80:20 were named T2N8, T5N5 and T8N2, respectively. The compacts with titanium/nickel and a weight ratio of 20:80 were sintered at 850 °C, 950 °C and 1050 °C, and were named T2N8-850, T2N8-950 and T2N8-1050, respectively.

### 2.2. Microstructural Characteriszation

The titanium nitride/nickel composite microstructures were characterized by a high-resolution field emission scanning electron microscopy (FESEM, JEOL JSM–6330TF, Tokyo, Japan). The compositions of the composites were identified by an energy dispersive X-ray microanalyzer (EDX, JEOL, Tokyo, Japan) equipped to the SEM. The crystalline structures of the composites were assessed by an X-ray diffraction (XRD, SIEMENS D5000, Munich, Germany) analysis with Cu K_α_ radiation.

### 2.3. Mechanical Properties Tests

The porosity of the composites was evaluated by the Archimedes’ technique. The microhardness of the composites was measured by Vickers indentations (Micro-Vickers Hardness Meter–Mitutoyo_MVK–H1, Kanagawa, Japan). Indentations were made on a perpendicular polished surface using a load of 50 g. The indentations were then examined by a light microscopy (LM, OLYMPUS, Tokyo, Japan). Additionally, the HRA hardness of the composite material was measured at three different locations on the surface using a Rockwell hardness tester and load of 60 kgf (Akashi Corporation AR-10, Hyogo, Japan). The wear test was performed with a ball-on-disk wear tester (Falex Tribology, Sugar Grove, IL, USA). Dry grinding processes with a 1 kg load in chromium steel ball and 200 rpm rotation speed were performed on the surfaces of the specimens at 22, 44, 66, 88 and 110 m. After the abrasion stopped, the weight of the specimen was measured for weight loss and the wear tracks were observed by LM. The compressive strength test was examined by a universal tester (JANOME JP-5004, Tokyo, Japan), and the compression rate was 1 mm/min.

## 3. Results and Discussion

### 3.1. Microstructure

Figure 1a–c show the microstructures of T2N8-850, T5N5-850 and T8N2-850, respectively. Dark particles in sphere shapes dispersed in a matrix were observed, and T8N2-850 had the highest amounts of the dark particles among other specimens. In Figure 1d, the high magnification view of Figure 1c, the nickel matrix (site a), the titanium nitride particle (site e) and the interface layer (site c) are shown. The EDS results in Table 1 show that the interface layer with a high concentration of oxygen and titanium was named as the oxygen-rich film. Parts of the Ti-oxide were dispersed in the matrix, as shown in site b. The layer (site d) with a thickness of approximately 3 μm and a high content of titanium, oxygen and nitrogen was named the nitrogen-rich layer.

The boundary of the titanium nitride particle (site d) in Figure 1d shows that oxidation and nitridation reactions occur on the titanium particles in a high-temperature sintering process under atmospheric conditions. The free energy of the titanium and oxygen reaction is lower than that of the titanium and nitrogen reaction [25]; therefore, the oxidation reaction takes precedence over the nitridation reaction on the titanium surface. However, the water-based hot forging lubricant covering the titanium particles helped prevent the titanium from making contact with the oxygen. Additionally, the carbon content decreased significantly in the nitrogen-rich layer according to the following carbothermal reduction reaction:2TiO_2_ + 4C + N_2_ → 2TiN + 4CO↑(1)

The carbothermal reduction reaction converts a portion of the titanium oxide to titanium nitride. Simultaneously, the carbon monoxide gas released from the interface breaks the oxygen-rich film. Consequently, the oxygen-rich film fragments are dispersed into the matrix around the titanium nitride particles, as shown at site b in Figure 1d.

In order to investigate the effect of temperature on the carbothermal reduction reaction, the thicknesses of the nitrogen-rich layer and the oxygen-rich film covering the outer surface of the titanium nitride particles were measured by SEM. The average thickness, shown in Figure 2, was calculated by measuring at least seven sites. The thickness of the nitrogen-rich layer increased as the sintering temperature increased (Figure 2a). In addition, the thickness of the oxygen-rich film depends not only on the temperature but also on the proportion of titanium in the composites (Figure 2b). Before sintering, the green compacts will be soaked in the water-based hot forging lubricant solution. The solution with C and O elements can penetrate into the interior of the green compacts through the gaps between the powders. Compared with T5N5 and T8N2, T2N8 has the smallest numbers of titanium particles. For T2N8, each titanium particle obtains relatively high amounts of the water-based hot forging lubricant solution to cover the surface when compared with T5N5 and T8N2; therefore, T2N8 has the thickest oxygen-rich film after sintering at 1050 °C (Figure 2b). However, for T5N5, as the sintering temperature increases to 1050 °C, the carbothermal reduction reaction intensifies to reduce the thickness of the oxygen-rich film and increase the thickness of the nitrogen-rich layer.

Figure 3 shows the SEM micrographs of T2N5, T5N5 and T8N2 sintered at 850, 950 and 1050 °C. In the T2N8-850, titanium nitride particles were easily detached from the nickel matrix by specimen polishing as indicated by the arrow in Figure 3a. Most of the titanium nitride particles remained in the nickel matrix because of the thick oxygen-rich film that bound the particles and the matrix of the T2N8-950. However, some pores appeared in the nitrogen-rich layer of the titanium nitride particles, as shown in Figure 3b. In Figure 3c, the titanium nitride particles remained in the nickel matrix, but fractures were observed within the titanium nitride particles, indicating that oxygen-rich film bonds well with the nickel matrix of the T2N8-1050, but that the numerous defects in the titanium nitride particles fracture the titanium nitride particles when the specimen is polished. Figure 3d–f shows a similar situation for the T5N5 sintered at 850, 950 and 1050 °C, respectively.

For T8N2-850 (Figure 3g), the detachment of the titanium nitride particles is more frequent than for T2N8-850, and parts of the titanium nitride particles inter-diffuse with each other in T8N2 sintered at 950 and 1050 °C, as shown in Figure 3h,i, respectively. The detachment of the nickel matrix was observed in Figure 3i after polishing, as titanium nitride is harder than the nickel. 

### 3.2. Composition

Figure 4 show XRD spectra of the T2N8, T5N5 and T8N2 compacts sintered at 850, 950 and 1050 °C. As shown in Figure 4a, the Ni phase in the XRD pattern of T2N8-850 is the main phase, and the intensity of the TiN_0.3_ diffraction peak is very weak. As the sintering temperature was further increased to 1050 °C, the intensity of the TiN_0.3_ diffraction peak decreased, and the TiN diffraction peak appeared. When the sintering temperature reached 1050 °C, the intensity of the diffraction peaks of both TiN and TiO_2_ significantly increased. An SEM-EDS analysis showed that titanium oxide was formed at 1050 °C. However, the titanium content of T2N8 is low, and the titanium oxide forms only on the outer surface of the titanium nitride particles, which results in low titanium oxide content. Therefore, titanium oxide cannot be detected in the XRD pattern of T2N8-850.

The XRD spectra of T5N5 in Figure 4b show that TiN_0.3_ was formed at 850 °C. At 950 °C, titanium is transformed into TiO_2_ and TiN. As the sintering temperature increased to 1050 °C, the diffraction peaks of TiN increased, while the diffraction peak of TiO_2_ decreased, suggesting that the thickness of the nitrogen-rich layer increases with sintering temperature because of violent carbothermal reduction reactions at 1050 °C. Figure 4c shows that the diffraction peaks of the TiO_2_ and TiN phases appear at 850, 950 and 1050 °C. Additionally, the intensity of the TiO_2_, TiN_0.3_ and TiN diffraction peaks of T8N2 is greater than that of said peaks in T2N8 and T5N5.

### 3.3. Porosity

Figure 5a,b shows the porosity and the density of pure nickel, T2N8, T5N5, and T8N2. As the sintering temperature increased from 850 to 950 °C, the porosity of the pure nickel decreased from 9.02% to 2.01%. The porosity of T2N8-950 is lower than that of T2N8-850 because the pores are eliminated by the thermal diffusion of the nickel powders. However, the carbothermal reduction reaction occurs violently at high temperatures (1050 °C) and forms numerous pores. Hence, the porosity of T2N8 increased to 10.43% when the T2N8 compact was sintered at 1050 °C. The porosity of the sintered T5N5 and T8N2 compacts increased with the increased sintering temperature, indicating that the degree of pore formation is greater than that of compact shrinkage from diffusion. This is because the high titanium content in the compacts fuels the carbothermal reduction reaction, which produces numerous pores.

The porosity of T5N5 is lower than that of T2N8 and T8N2 sintered at 850, 950, and 1050 °C. According to the literature, a wider powder particle distribution range, a stable sintered compact shrinkage rate and smaller pores produce a more uniform crystal phase distribution [26]. In this study, the average particle sizes of titanium and nickel powders are 75 and 45 μm, respectively. When the weight ratio of these two powders is equivalent, a larger particle size distribution range is obtained; therefore, the porosity of Ti5N5 is lower than that of T2N8 or T8N2.

### 3.4. Hardness

Figure 6a shows the Rockwell hardness of T2N8, T5N5 and T8N2 sintered at 850, 950, and 1050 °C. The hardness increased with an increased sintering temperature and number of titanium nitride particles. Figure 6b shows the hardness of the center of the T2N8 titanium nitride particles as measured by the Vickers microhardness tester. The center of the T2N8 titanium nitride particles increased from approximately 700 HV to 1100 HV when the sintering temperature rose from 859 to 1050 °C. The hardness of the composites is greater than that of pure titanium (150 HV) and nickel (123.6 HV) [27,28]. Obviously, the transformation of titanium into TiN and TiN_0.3_ reinforces the composites. The hardness of titanium nitride reported by previous studies is 900 HV [29], 1300 HV [30] or 2000 HV [31], which depends on the titanium to nitrogen ratio. Not only the hardness of particles and matrix, but also the adhesion of particles and matrix affects the hardness of the composites. T8N2 sintered at 1050 °C has thinner oxygen-rich layer than T8N2 sintered at 950 °C, which suggests the titanium nitride particles cannot be anchored well by the matrix. During the Rockwell hardness test, the titanium nitride particles are expected to easily detach form the matrix via the plastic deformation. This is why T8N2 sintered at 1050 °C has a lower Rockwell hardness than T8N2 sintered at 950 °C.

Figure 7 shows LM micrographs of the indentation locations of the Vicker’s microhardness of T2N8-950. The center of the titanium nitride particles is HV1033.0 in Figure 7a. The hardness of the oxygen-rich film is HV318.2, which is between that of the nickel matrix (HV79.7) and the titanium nitride particles (HV1033.0) in Figure 7b.

### 3.5. Wear Resistance

The factors affecting the wear resistance of a composite include (1) the number of reinforcing materials in the matrix, (2) the combination of reinforcing materials in the matrix, and (3) the porosity of the composite. In Figure 8a, T2N8 sintered at 850 and 950 °C has the highest and lowest weight loss from wear, respectively. The thickness of the oxygen-rich film strengthens the bonds between the reinforcing material and the matrix. However, the porosity generated by the carbothermal reduction reaction increases rapidly at 1050 °C and lowers the wear resistance of T2N8-1050 more so than that of T2N8-950. T2N8-850 has the lowest wear resistance because its oxygen-rich film is very thin, and T2N8-850 has the lowest hardness among the T2N8 composites. T5N5 shows the same relationship between wear resistance and sintering temperature as does T2N8 (Figure 8b).

Figure 8c shows the wear resistance of T8N2 sintered at 850, 950 and 1050 °C. T8N2-1050 and T8N2-850 have the lowest and highest weight loss, respectively. The T8N2 matrix consists primarily of titanium nitride particles, some of which contact with each other. Therefore, the bonding ability of titanium nitride particles increases with the sintering temperature, leading to a significant reduction in weight loss. However, since the nickel content is low, the nickel powder is discontinuously distributed in the titanium matrix; accordingly, the nickel peels off easily, as shown in Figure 9. Furthermore, multiple carbothermal reduction reactions cause poor interfacial adhesion between the titanium and the nickel, and allow titanium nitride particles to be easily worn away. As such, T8N2 has the lowest wear resistance among the sintered composites, even through T8N2 has the highest hardness. The coefficients of friction for T2N8-950, T5N5-950 and T8N2-950 were 0.21, 0.36 and 0.81, respectively. T2N8-950 and T5N5-950 have similar coefficients of friction and wear-induced weight loss (Figure 8a,b). T8N2-950 possessed the highest coefficient of friction and the worst wear resistance among other groups. The high coefficient of friction for T8N2-950 can be attributed to the fact that the titanium nitride particles are easy to detach under wear tests, and the pits left on the specimen surface. Previous studies found that the hardness and the wear resistance of composites are not directly related. During wear processing, shear stress occurs on the composite, and cracks may appear in the matrix [32]. As a result, even if the hardness of the composite is enhanced by particle reinforcement, the reinforced particles may be easily peeled off from internal cracks.

### 3.6. Compressive Strength

Figure 10 show the compressive stress–strain curves of T2N8, T5N5, and T8N2 sintered at 850, 950, and 1050 °C. A sintering temperature of 950 °C provides a higher compressive strength for T2N8 and T5N5 than does 1050 °C, as shown in Figure 10a,b. The worse compressive strength is from the high porosity of the compacts sintered at 1050 °C. The position around the pores is likely to cause stress concentration and become the starting point of deformation. The compressive strength of T2N8 and T5N5 sintered at 1050 °C is higher than that of T2N8 and T5N5 sintered at 850 °C. This result is consistent with the wear test. Furthermore, T8N2 exhibited the lowest compressive strength of the sintered compacts (Figure 10c), and shear formation on T8N2 was observed. This is because T8N2 contains high amounts of hard material (titanium nitride) and low amounts of soft material (nickel). This combination produces brittle fractures, which occur via the shear force induced when the T8N2 composites are under compressive loads.

## 4. Conclusions

This study successfully fabricated a titanium nitride/nickel composite through the carbothermal reduction reaction, which was achieved by sintering the compacts of titanium and nickel that were mixed with a water-based hot forging lubricant. The effects of the titanium powder ratios (20, 50 and 80 wt.%) and sintering temperatures (850, 950 and 1050 °C) on the mechanical properties of the composites were investigated. The composites with titanium–nickel weight ratios of 20:80 sintered at 950 °C exhibited the highest wear resistance (~7 × 10^−4^ g/cm^2^ weight loss after 110 m wear tests) and compressive strength (~330 MPa) among other groups. A titanium-oxide film was observed to form around the titanium nitride and provided good bonding with the nickel matrix. A violent carbothermal reduction reaction is induced at high temperatures (1050 °C) and generates abundant pores in the composites, which deteriorate the wear resistance and the compressive strength.

## Figures and Tables

**Figure 1 materials-13-04473-f001:**
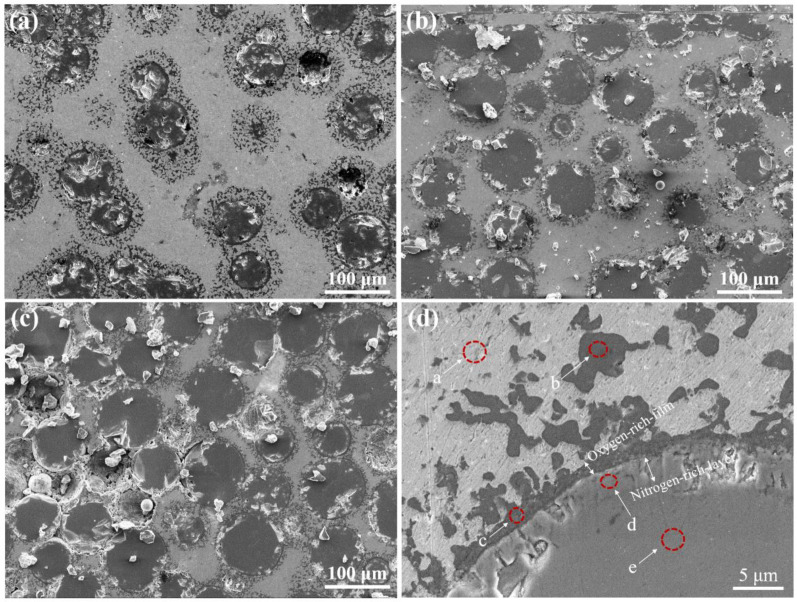
SEM micrographs of (**a**) T2N8-850, (**b**) T5N5-850, (**c**) T8N2-850. (**d**) A high magnification view of (**c**).

**Figure 2 materials-13-04473-f002:**
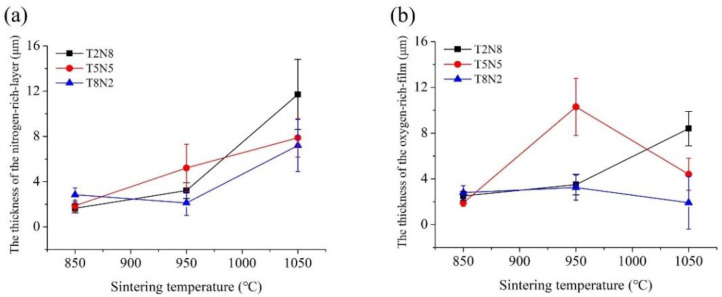
(**a**) Thickness of the nitrogen-rich layer of the titanium nitride particles. (**b**) Thickness of the oxygen-rich film between the titanium nitride particles and the nickel matrix.

**Figure 3 materials-13-04473-f003:**
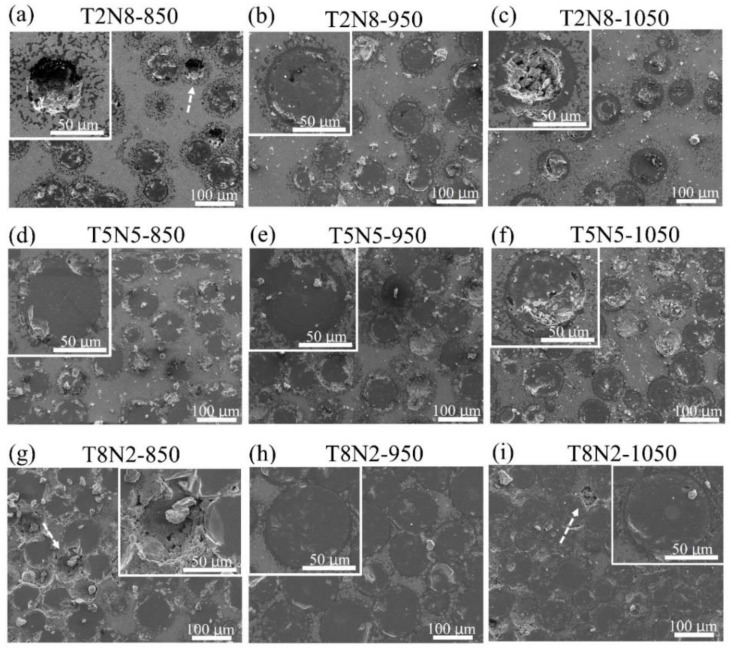
SEM micrographs of T2N5, T5N5 and T8N2 sintered at various temperatures. (**a**) T2N8-850, (**b**) T2N8-950, (**c**) T2N8-1050, (**d**) T5N5-850, (**e**) T5N5-950, (**f**) T5N5-1050, (**g**) T8N2-850, (**h**) T8N2-950 and (**i**) T8N2-1050.

**Figure 4 materials-13-04473-f004:**
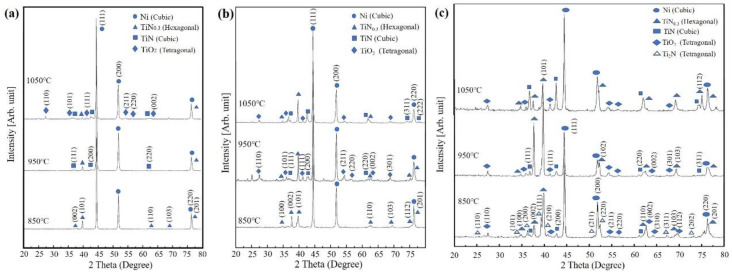
XRD spectra of the (**a**) T2N8, (**b**) T5N5, and (**c**) T8N2 compact sintered at various temperatures.

**Figure 5 materials-13-04473-f005:**
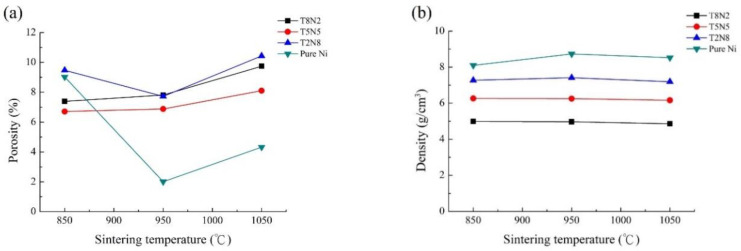
(**a**) The porosity and (**b**) the density of pure nickel, T2N8, T5N5, and T8N2 sintered at various temperatures.

**Figure 6 materials-13-04473-f006:**
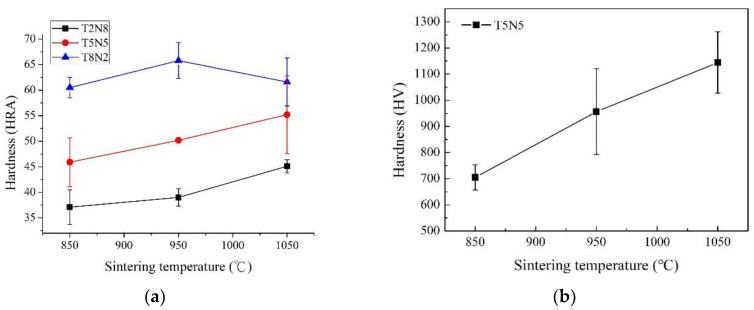
(**a**) Rockwell hardness of T2N8, T5N5 and T8N2 sintered at various temperatures. (**b**) Vickers microhardness of T2N8. Indentation is at the center of the titanium nitride particles sintered at various temperatures.

**Figure 7 materials-13-04473-f007:**
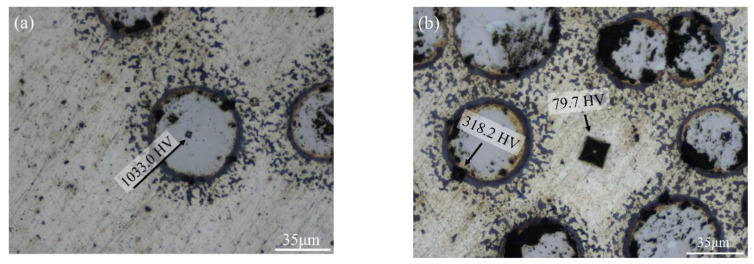
(**a**) LM micrograph of the indentation location in center of the T2N8-950 titanium nitride particles. (**b**) LM micrograph of the indentation location in the T2N8-950 nickel matrix and oxygen-rich film.

**Figure 8 materials-13-04473-f008:**
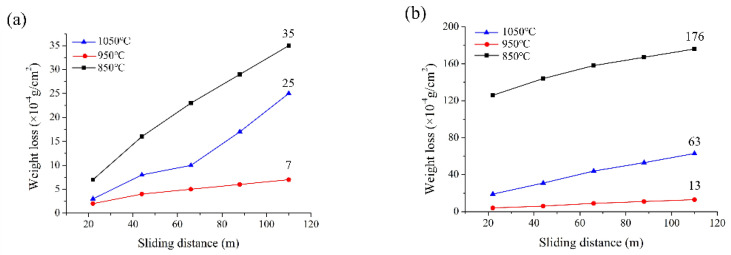
Weight losses of (**a**) T2N8, (**b**) T5N5 and (**c**) T8N2 sintered at various temperatures after wear tests.

**Figure 9 materials-13-04473-f009:**
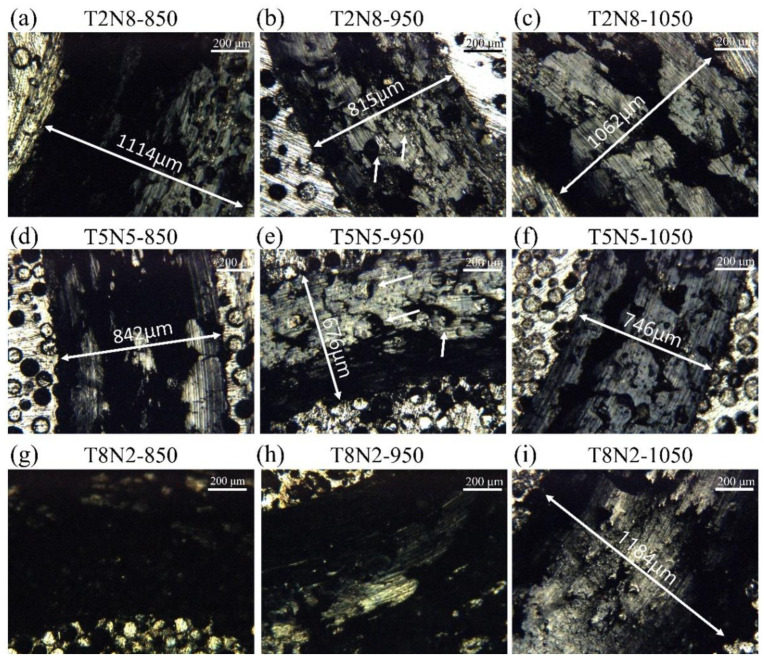
LM micrographs of T2N5, T5N5, and T8N2 after wear test. (**a**) T2N8-850, (**b**) T2N8-950, (**c**) T2N8-1050, (**d**) T5N5-850, (**e**) T5N5-950, (**f**) T5N5-1050, (**g**) T8N2-850, (**h**) T8N2-950, and (**i**) T8N2-1050.

**Figure 10 materials-13-04473-f010:**
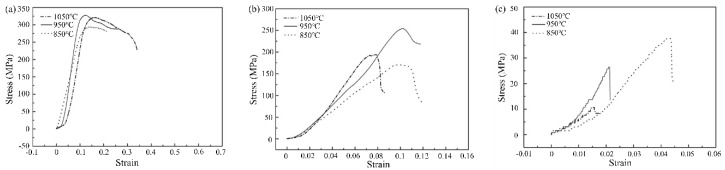
Stress–strain curves of (**a**) T2N8, (**b**) T5N5, and (**c**) T8N2 sintered at various temperatures under compressive tests.

**Table 1 materials-13-04473-t001:** Chemical composition of T8N2-850 (Figure 1d) detected by EDS. The detected areas a, b, c, d and e are marked in Figure 1.

Element (at %)	Site a	Site b	Site c	Site d	Site e
Carbon	8.2	1.7	4.2	0.7	5.1
Nitrogen	0	0	0	28.4	5.9
Oxygen	7.5	72.0	67.3	27.9	31.7
Titanium	2.7	25.0	25.5	42.9	57.3
Nickel	81.6	1.3	3.0	0.1	0

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
