# Peer review of "Effects of Reinforcement Ratios and Sintering Temperatures on the Mechanical Properties of Titanium Nitride/Nickel Composites"

_materials, 2020, doi:10.3390/ma13204473_

Round 1

Reviewer 1 Report

This work is a systematic study understanding the effects of sintering temperature and various ratio of the titanium nitride and nickel composites. However, the introduction was severely lacking for foundation of the study and I found major issues with the clarity of the study. Without addressing the reason for choosing this material, this study does not really have any significance to readers. The quality of presentation is also rather poor as many figures are non-legible. Please see my comments for the study.

  • The introduction severely lacked of motivation of why titanium nitride and nickel were chosen as the particle and matrix. I think it is important to explain why these materials were chosen, and if there were any previous work done on this system.
  • There is a very large confusion in this study, as the authors refer it to a titanium nitride and nickel composite. The authors repeatedly refer to “Ti particle” throughout the manuscript, but it is supposed to be Ti nitride. This needs to be clarified throughout. There was no Ti peak observed in XRD, and the SEM/EDS even showed low concentrations of N in the Ti particle. If it is just Ti, then it should be a Ti/TiN/Ni composite.
  • The overall organization of the presented data could be improved. Since the study is focused on the effects of sintering temperature and ratio, those results should be presented first. For example, Figure 3 should be the first figure introduced, before Figure 1 and Figure 2 which show more detailed microstructure of a selected sample.
  • The error bars for the various thickness is very large just for 7 different sites, especially for T2N8 and was not explained. Could a larger statistical analysis improve the size of these scale bars? Also, only the increase in thickness was explained in the discussion, and T5N5 showed a completely different trend from the other samples, which was not discussed.
  • The hardness was compared to pure titanium and nickel, which is clearly much lower than titanium nitride. However, your matrix composite in this study consists of titanium nitride, not pure titanium. The hardness of titanium nitride should be included here.
  • Are there literature values to compare for Rockwell hardness to titanium nitride and Ni to show how your results compare? Based on the variations in microhardness from HV1033 to HV79 in various locations, how could an average microhardness be obtained? It would be too dependent on the location of where the indentation was performed. You would need a much larger sample size for proper statistical analysis.
  • Figure 1(b)-1(f) are too small for viewing. They can be removed since they are not very useful.
  • High magnification SEM/EDS should also be presented for the T5-5N and T2-8N samples, and would be beneficial for side by side with Figure 1(a).
  • Figure 3 – the smaller insets should have some border around and another scale bar. They need a border because they are almost camouflaging with the background and it is very hard to see. The scale bar is impossible to see, even after magnifying in.
  • Table 1 – headers should be site a, site b…site e, instead of just “a,b,c,d,e”
  • Figures 9 and 10 axis labels and legends are too small for viewing.

Author Response

  1. The introduction severely lacked of motivation of why titanium nitride and nickel were chosen as the particle and matrix. I think it is important to explain why these materials were chosen, and if there were any previous work done on this system.

Reply: Thanks for the suggestion. The choice of materials and their relative references have been added in the introduction.

Ni and its alloys are widely used in engineering applications in corrosion environments, such as chemical plant and nuclear power plant due to their strength, corrosion, and wear resistance at high temperatures. Such excellent properties make Ni is of great interest for the choice of components in the aggressive environments. In order to enlarge the range of Ni alloys’ applications in engineering fields, the Ni matrix composites produced by addition of metal oxides, carbides, and nitrides into the Ni matrix have been developed [A10–A11]. TiN has high melting point (2927 C) with high hardness, corrosion resistance and good thermal stability [A12]. TiN has been used in tribological applications in different forms, i.e., thin film on machining tools [A13,14] and reinforced particles in composites [A15].

Ibrahim et al. produced Ni-TiN composites via a DC electrodeposition method. The results indicated that the TiN microparticles in the nickel matrix greatly improved the corrosion resistance in a 3.5% NaCl solution and possessed higher hardness than a pure Ni [A15]. Ramesh Bapu et al. have synthesized Ni–titanium carbo nitride composites by electrodeposition and evaluated the corrosion resistances of the composites. It was found that the Ni–titanium carbo nitride composites showed better corrosion resistance in 3.5 wt.% NaCl solution and higher hardness and better wear resistance than nickel. The degree of the improvement depends on the grain size and the volume percent of titanium carbo nitride in the composites [A16].

[A10] C. Buelens, J. Fransaer, J.P. Celis, J.R. Roos, The mechanism of electrolytic codeposition of particles with metals. A review. Bull. Electrochem. 8 (1992) 371 - 375.

[A11] R.V. Williams, P.W. Martin, Electrodeposited Composite Coatings. Trans. Inst. Met. Finish. 42 (1964) 182-188.

[A12] H. Itoh, K. Kato and K. Sugiyama, Chemical vapour deposition of corrosion-resistant TiN film to the inner walls of long steel tubes. J. Mater. Sci., 21 (1986) 751-756.

[A13] S. Zhang, Material Development of Titanium Carbonitride-Based Cermets for Machining Application. Key. Eng. Mater. 138 (40) (1998) 521-544.

[A14] H. Zhang, S. Tang, J. Yan, X. Hu, Cutting performance of titanium carbonitride cermet tools. Int. J. Refract. Met. Hard. Mater. 25 (2007) 440-444.

[A15] Magdy A.M. Ibrahim, F. Kooli, Saleh N. Alamri, Electrodeposition and Characterization of Nickel–TiN Microcomposite Coatings. Int. J. Electrochem. Sci., 8 (2013) 12308 – 12320.

[A16] G.N.K. Ramesh Bapu, Sobha Jayakrishnan, Development and characterization of electro deposited Nickel–Titanium CarboNitride (TiCN) metal matrix nanocomposite deposits. Surf. Coat. Tech. 206 (2012) 2330–2336.

  1. There is a very large confusion in this study, as the authors refer it to a titanium nitride and nickel composite. The authors repeatedly refer to “Ti particle” throughout the manuscript, but it is supposed to be Ti nitride. This needs to be clarified throughout. There was no Ti peak observed in XRD, and the SEM/EDS even showed low concentrations of N in the Ti particle. If it is just Ti, then it should be a Ti/TiN/Ni composite.

Reply: Thanks for suggestion. The “titanium particle” has been replaced by “titanium nitride”. Please see the revised manuscript.

  1. The overall organization of the presented data could be improved. Since the study is focused on the effects of sintering temperature and ratio, those results should be presented first. For example, Figure 3 should be the first figure introduced, before Figure 1 and Figure 2 which show more detailed microstructure of a selected sample.

Reply: Figure 3 in the original manuscript has moved to Figure 1. In addition, we added SEM images in low magnification view in Figure.1. Please see the revised manuscript.

  1. The error bars for the various thickness is very large just for 7 different sites, especially for T2N8 and was not explained. Could a larger statistical analysis improve the size of these scale bars? Also, only the increase in thickness was explained in the discussion, and T5N5 showed a completely different trend from the other samples, which was not discussed.

Reply: In the revised manuscript, the titanium particles with similar size were chosen for measuring the thickness. Therefore, the error bar in the curve in the revised manuscript is smaller than that in the original manuscript. Please see the revised manuscript. For T2N8, each titanium particle obtains relatively higher amounts of the water-based hot forging lubricant solution to cover the surface when comparing with T5N5; therefore, T2N8 has a thickest oxygen-rich-film during sintering at 1050℃. However, for T5N5, as the sintering temperature increased to 1050℃, the carbothermal reduction reaction intensifies to reduce the thickness of the oxygen-rich-film.

  1. The hardness was compared to pure titanium and nickel, which is clearly much lower than titanium nitride. However, your matrix composite in this study consists of titanium nitride, not pure titanium. The hardness of titanium nitride should be included here.

Reply: The hardness of titanium nitride has been added into the content of manuscript.

Hardness of titanium nitride reported by previous studies is 900 HV[1], 1300 HV [2], 2000HV[3] which depends on Ti:N ratio. Please see the revised manuscript.

[R1] Bartłomiej, J.; Leszek, K. Nitriding of titanium and Ti6Al4V alloy in ammonia gas under low pressure. Mater. Sci. Technol. 2010, 26, 586-590.

[R2] Tamura, Y.; Yokoyama, A.; Watari, F. Uo, M.; Kawasaki, T. Mechanical Properties of Surface Nitrided Titanium for Abrasion Resistant Implant Materials. Mater. Trans. 2002, 4, 3043-3051

[R3] Ruud P. van Hove; Inger N. Sierevelt, Barend J. van Royen, Peter A. Nolte, Titanium-Nitride Coating of Orthopaedic Implants: A Review of the Literature. Biomed Res. Int. 2015, 1-9.

  1. Are there literature values to compare for Rockwell hardness to titanium nitride and Ni to show how your results compare? Based on the variations in microhardness from HV1033 to HV79 in various locations, how could an average microhardness be obtained? It would be too dependent on the location of where the indentation was performed. You would need a much larger sample size for proper statistical analysis.

Reply: The Rockwell hardness of the compacts were added in Figure 6 in the revised manuscript.

  1. Figure 1(b)-1(f) are too small for viewing. They can be removed since they are not very useful.

Reply: Figure 1(b)-1(f) has been removed. Please see the revised manuscript.

  1. High magnification SEM/EDS should also be presented for the T5-5N and T2-8N samples, and would be beneficial for side by side with Figure 1(a).

Reply: The SEM image of T5-5N and T2-8N samples has been added in Figure.1. Please see the revised manuscript.

  1. Figure 3 – the smaller insets should have some border around and another scale bar. They need a border because they are almost camouflaging with the background and it is very hard to see. The scale bar is impossible to see, even after magnifying in.

Reply: The images has been revised according to the reviewer’s suggestion. Please see the revised manuscript.

  1. Table 1 – headers should be site a, site b…site e, instead of just “a,b,c,d,e”

Reply: The table has been revised according to the reviewer’s suggestion.

Element (at%)

Site a

Site b

Site c

Site d

Site e

C

8.2

1.7

4.2

0.7

5.1

N

0

0

0

28.4

5.9

O

7.5

72.0

67.3

27.9

31.7

Ti

2.7

25.0

25.5

42.9

57.3

Ni

81.6

1.3

3.0

0.1

0

  1. Figures 9 and 10 axis labels and legends are too small for viewing.

Reply: The axis labels and legends have been enlarged. Please see the revised manuscript.

Reviewer 2 Report

Dear Authors,

I have read your manuscript carefully and I would say that it would be interesting for readers. The objectives of the study are clearly defined. The introduction provides a good, generalized background of the topic. The results are sufficiently explained and are presented in an appropriate format. The figures show essential data; some of the data are also summarized in the text. The cited literature is relevant to the study and balanced. I have only a few comments which will improve the manuscript as well enable more complementary explanation of the presented results.

What was the reason to make the sintering in air atmosphere?

The purity of the powders and the morphology of the powder particles need to be presented.

Not all the peaks were identified in the XRD spectra presented in Figures 4-6. Identify the unknown phases in detail, please.

I recommend converting Table 2 into a figure. It will be more readable.

Please explain why the hardness of T8N2 decreased with increasing the sintering temperature up to 1050°C. This do not correspond with other results of T5N5 and T2N8.

I recommend using “LM micrograph”, not “OM image” in the description of Figure 8.

Why did the authors not present the specific wear rate (SWR) but only weight loss as a result of the wear tests? What was the material of the counter specimen (ball)? What was the coefficient of friction (CoF)?

The conclusions need to be revised by the authors. There is a lack of values of the mechanical properties obtained using the optimal consolidation parameters.

In general, the methodology is lacking in information.

A general remark is that the authors did not use the plural of some words where it is needed, e.g. micrograph instead of micrographs (see Figure 3).

Author Response

  1. What was the reason to make the sintering in air atmosphere?

Reply: The Ti will be transferred into TiN according to the following equation. In the reaction, TiO2 is the precursor and it is formed by oxidation of Ti. Hence, the sintering shall be in air atmosphere.

2TiO2 + 4C + N2 → 2TiN + 4CO↑   

  1. The purity of the powders and the morphology of the powder particles need to be presented.

Reply: Commercial titanium (>99.5%) and nickel (>99.5%) powders with average diameters of 75 and 45 μm in spherical shape, respectively, were used. The description has been added in the revised manuscript.

  1. Not all the peaks were identified in the XRD spectra presented in Figures 4-6. Identify the unknown phases in detail, please.

Reply: The peaks in XRD spectra has been identified.

  1. I recommend converting Table 2 into a figure. It will be more readable.

Reply: The Table 2 has been converted into Figure 5 in the revised manuscript.

  1. Please explain why the hardness of T8N2 decreased with increasing the sintering temperature up to 1050°C. This do not correspond with other results of T5N5 and T2N8.

Reply: Figure 6(a) shows Rockwell hardness of T8N2 sintered at 850, 950, and 1050 ℃. Not only the hardness of particles and matrix but also the adhesion of particles and matrix affect the hardness of the composites. For T8N2 sintered at 1050 ℃, it has thinner oxygen-rich layer than T8N2 sintered at 950 ℃, which suggest the TiN particles can not be anchored well by Ni matrix. During Rockwell hardness test, the plastic deformation cause TiN particles detach form the Ni matrix. It is why T8N2 sintered at 1050 ℃ has a lower Rockwell hardness than T8N2 sintered at 950 ℃. Wear tests also demonstrated this result.

  1. I recommend using “LM micrograph”, not “OM image” in the description of Figure 8.

Reply: The OM has been replaced by LM in the revised manuscript.

  1. Why did the authors not present the specific wear rate (SWR) but only weight loss as a result of the wear tests? What was the material of the counter specimen (ball)? What was the coefficient of friction (CoF)?

Reply: The counter specimen is chrome steel ball. The coefficient of friction for T2N8-950, T5N5-950, and T8N2-950 was 1.59, 0.39, and 0.81, respectively. Obvious, the higher amounts of the titanium nitride particles in T5N5-950 than T2N8-950, which provides lower coefficient of friction. For T8N2, the titanium nitride particles are easily to detach under wear and the pits left on the specimen surface increases the coefficient of friction.

  1. The conclusions need to be revised by the authors. There is a lack of values of the mechanical properties obtained using the optimal consolidation parameters.

Reply: The conclusion has been revised. The mechanical properties of the composite made by the optimal parameters has been added. The PMMCs with titanium-nickel weight ratios of 20:80 sintered at 950 ℃ exhibited the highest wear resistance (~ 7 x 10-4 g/cm2 weight loss after 110 m wear tests) and compressive strength (~330 MPa) among other groups. The description has been added in the conclusion.

  1. In general, the methodology is lacking in information.

Reply: the methodology has been in the material and method. The mechanical properties of the PMMCs strongly depend on the weight ratio of reinforcement particles and matrix materials. In this study, three weight ratios of reinforcement particles and matrix materials were chosen to find the optimal parameter in terms of wear resistance and compressive strength of the PMMCs. Furthermore, the sintering temperature significantly affects the density of the PMMCs and, therefore, three sintering temperatures were applied.

A general remark is that the authors did not use the plural of some words where it is needed, e.g. micrograph instead of micrographs (see Figure 3).

  Reply: Some words need to be plural have been revised. Please see the revised manuscript.

Reviewer 3 Report

In this work, titanium nitride/nickel metal matrix composites has been produced via powder metallurgy. Thereafter, The effects of the amount of titanium powder and the sintering temperature on the mechanical properties (hardness, wear resistance, and compressive strength) of the composites have been evaluated. Before further consideration the following issues should be considered and addressed:

  1. The novelty aspect is missing and should be explained in the last paragraph of the introduction section.
  2. An state of art as regards the Ni matrix composites should be included in the introduction.
  3. Page 2 line 73, instead of SEM should be FESEM.
  4. A low magnification image should be inserted to show the dispersion of the reinforcement.
  5. The EDS spectrums in Fig, 1 are not readable. Please modify them to make the more informative., otherwise remove them.
  6. Have the results of the density and porosity been confirmed by the image analysis method?
  7. The graphs in figure 9, and 10 are not so clear, please modify them to make them readable.
  8. Has any SEM analysis been carried out on the samples after the wear test to clearly show the differences?
  9. Has any sign of shear formation in the compression test of composites been revealed? If yes, why?
  10. What could be the dominant strengthening mechanism in the composite? Why?
  11. Is the results of this work comparable with the literature?

Author Response

  1. The novelty aspect is missing and should be explained in the last paragraph of the introduction section.

Reply: The introduction has been revised and the novelty aspect was added.

The production of PMMCs with ceramic particles can be easily achieved by powder metallurgy. However, the cost of ceramic powder production is very high. This study aimed to use a carbothermal reduction reaction to transform metal particles into ceramic particles during sintering.

  1. A state of art as regards the Ni matrix composites should be included in the introduction.

Reply: The introduction has been revised.

Ni and its alloys are widely used in engineering applications in corrosion environments, such as chemical plant and nuclear power plant due to their strength, corrosion, and wear resistance at high temperatures. Such excellent properties make Ni is of great interest for the choice of components in the aggressive environments. In order to enlarge the range of Ni alloys’ applications in engineering fields, the Ni matrix composites produced by addition of metal oxides, carbides, and nitrides into the Ni matrix have been developed [A10–A11]. TiN has high melting point (2927 C) with high hardness, corrosion resistance and good thermal stability [A12]. TiN has been used in tribological applications in different forms, i.e., thin film on machining tools [A13,14] and reinforced particles in composites [A15].

Ibrahim et al. produced Ni-TiN composites via a DC electrodeposition method. The results indicated that the TiN microparticles in the nickel matrix greatly improved the corrosion resistance in a 3.5% NaCl solution and possessed higher hardness than a pure Ni [A15]. Ramesh Bapu et al. have synthesized Ni–titanium carbo nitride composites by electrodeposition and evaluated the corrosion resistances of the composites. It was found that the Ni–titanium carbo nitride composites showed better corrosion resistance in 3.5 wt.% NaCl solution and higher hardness and better wear resistance than nickel. The degree of the improvement depends on the grain size and the volume percent of titanium carbo nitride in the composites [A16].

[A10] C. Buelens, J. Fransaer, J.P. Celis, J.R. Roos, The mechanism of electrolytic codeposition of particles with metals. A review. Bull. Electrochem. 8 (1992) 371 - 375.

[A11] R.V. Williams, P.W. Martin, Electrodeposited Composite Coatings. Trans. Inst. Met. Finish. 42 (1964) 182-188.

[A12] H. Itoh, K. Kato and K. Sugiyama, Chemical vapour deposition of corrosion-resistant TiN film to the inner walls of long steel tubes. J. Mater. Sci., 21 (1986) 751-756.

[A13] S. Zhang, Material Development of Titanium Carbonitride-Based Cermets for Machining Application. Key. Eng. Mater. 138 (40) (1998) 521-544.

[A14] H. Zhang, S. Tang, J. Yan, X. Hu, Cutting performance of titanium carbonitride cermet tools. Int. J. Refract. Met. Hard. Mater. 25 (2007) 440-444.

[A15] Magdy A.M. Ibrahim, F. Kooli, Saleh N. Alamri, Electrodeposition and Characterization of Nickel–TiN Microcomposite Coatings. Int. J. Electrochem. Sci., 8 (2013) 12308 – 12320.

[A16] G.N.K. Ramesh Bapu, Sobha Jayakrishnan, Development and characterization of electro deposited Nickel–Titanium CarboNitride (TiCN) metal matrix nanocomposite deposits. Surf. Coat. Tech. 206 (2012) 2330–2336.

  1. Page 2 line 73, instead of SEM should be FESEM.

Reply: The SEM has been replaced by FESEM.

  1. A low magnification image should be inserted to show the dispersion of the reinforcement.

Reply: Low magnification images of the sintered compacts are shown in Figure. 3.

  1. The EDS spectrums in Fig, 1 are not readable. Please modify them to make the more informative., otherwise remove them.

 Reply: The EDS spectrums are removed according to the suggestion.

  1. Have the results of the density and porosity been confirmed by the image analysis method?

The graphs in figure 9, and 10 are not so clear, please modify them to make them readable.

Reply: (1) The results of porosity are roughly consist with the SEM images. However, the images only cover some ranges of the compacts. Therefore, the porosity of the compact was evaluated by the Archimedes’ technique. (2) The labels and legends in graphs are enlarged.

  1. Has any SEM analysis been carried out on the samples after the wear test to clearly show the differences?

Reply: The images of the samples after the wear tests were shown in Figure 9 in the revised manuscript.

  1. Has any sign of shear formation in the compression test of composites been revealed? If yes, why?

Reply: Shear formation occurs on T8N2 which was under compressive loads. Therefore, T8N2 exhibited the lowest compressive strength of the sintered compacts. because T8N2 has high amounts of hard material (titanium nitride) and low amounts of soft material (nickel).

  1. What could be the dominant strengthening mechanism in the composite? Why?

Reply: The main strengthening mechanism is that hard particles dispersed in the relatively soft matrix. Furthermore, the oxygen-rich-films have hardness between TiN particles and Ni matrix. It makes the TiN particles can well bond with the matrix.

  1. Is the results of this work comparable with the literature?

Reply: The hardness of titanium nitride from the literature has been added into the content of manuscript to compared with the results of this work.

Figure 6(b) shows the centre of the T2N8 titanium nitride particles increased from approximately 700 HV to 1100 HV when the sintering temperature raised from 859 to 1050 ℃. In addition, hardness of titanium nitride reported by previous studies is 900 HV[1], 1300 HV [2], 2000HV[3] which depends on Ti:N ratio.

[R1] Bartłomiej, J.; Leszek, K. Nitriding of titanium and Ti6Al4V alloy in ammonia gas under low pressure. Mater. Sci. Technol. 2010, 26, 586-590.

[R2] Tamura, Y.; Yokoyama, A.; Watari, F. Uo, M.; Kawasaki, T. Mechanical Properties of Surface Nitrided Titanium for Abrasion Resistant Implant Materials. Mater. Trans. 2002, 4, 3043-3051

[R3] Ruud P. van Hove; Inger N. Sierevelt, Barend J. van Royen, Peter A. Nolte, Titanium-Nitride Coating of Orthopaedic Implants: A Review of the Literature. Biomed Res. Int. 2015, 1-9.

Round 2

Reviewer 1 Report

Thank you for the great effort in revising the manuscript. It has significantly improved the quality of the paper, especially the introduction. I just have a few minor comments which I found.

Please make sure all the figure captions are consistently in the same style. It is typically “Figure 1. Caption here”, not “Figure. 1 Caption here”.

Figure 2(a) the y axis scale can be reduced, to better see the changes in layer thickness.

XRD plot labels are non-legible due to low quality and small font sizes. Please try to get a higher quality image and increase font sizes.

I would recommend doing some proof reading throughout. Section 4 should be “Conclusions” and 5 should be “References”

Author Response

Comments and Suggestions for Authors

Thank you for the great effort in revising the manuscript. It has significantly improved the quality of the paper, especially the introduction. I just have a few minor comments which I found.

  1. Please make sure all the figure captions are consistently in the same style. It is typically “Figure 1. Caption here”, not “Figure. 1 Caption here”.

Reply: The figure captions were revised to be consistently. Please see the revised manuscript.

  1. Figure 2(a) the y axis scale can be reduced, to better see the changes in layer thickness.

Reply: The y axis scale in Figure 2 has been changed from 0-24 to 0-16. Please see Figure 2 in the revised manuscript.

  1. XRD plot labels are non-legible due to low quality and small font sizes. Please try to get a higher quality image and increase font sizes.

Reply: The XRD plot with high resolution and big font size has been attached in the revised manuscript.

  1. I would recommend doing some proof reading throughout. Section 4 should be “Conclusions” and 5 should be “References”.

Reply: We have revised the contents according to the recommendations. Please see Figure 2 in the revised manuscript.

Reviewer 2 Report

Dear Authors,

Thank you for taking into account the reviewer’s suggestions. The manuscript is significantly improved; however, a few new corrections and explanations are needed.

In Figure 2 please change the values of the y-axis from 0-24 to 0-16.

The authors should decide which term they would like to use: “micrograph” or “image”. Now, both forms are used. The reviewer suggests using the term “micrograph”.

The coefficient of friction (CoF) of T2N8-950 is 1.59. How is it possible? Look at results of CoF one more time, please. Deep explanation is needed.

Author Response

Dear Authors,

Thank you for taking into account the reviewer’s suggestions. The manuscript is significantly improved; however, a few new corrections and explanations are needed.

  1. In Figure 2 please change the values of the y-axis from 0-24 to 0-16.

Reply: The y axis scale in Figure 2 has been changed from 0-24 to 0-16. Please see Figure 2 in the revised manuscript.

  1. The authors should decide which term they would like to use: “micrograph” or “image”. Now, both forms are used. The reviewer suggests using the term “micrograph”.

Reply: The term “image” has been replaced by “micrograph” for consistency. Please see the revised manuscript.

  1. The coefficient of friction (CoF) of T2N8-950 is 1.59. How is it possible? Look at results of CoF one more time, please. Deep explanation is needed.

Reply: Thanks for the reminding. We inspected the specimens and found the surface roughness of the specimens were not the same. Therefore, we machined the specimen surfaces by milling machine and tested the CoF of the specimens again. The coefficient of friction for T2N8-950, T5N5-950, and T8N2-950 was 0.21, 0.36, and 0.81, respectively. T2N8-950 and T5N5-950 have similar coefficient of friction and wear-induced weight loss (Figure 8(a) and (b)). For T8N2-950, it possessed highest coefficient of friction and the worst wear resistance among other groups. The high coefficient of friction for T8N2-950 can be attributed that the titanium nitride particles are easily to detach under wear tests and the pits left on the specimen surface.

Reviewer 3 Report

The revision is satisfactory and so the paper can be accepted in this form. 

Author Response

  1. The revision is satisfactory and so the paper can be accepted in this form. 

Reply: Thanks for the recommendations. The manuscript still be minor revised according to other reviewers’ suggestions. Please see the revised manuscript.
